# SceneCraft: Layout-Guided 3D Scene Generation

**Xiuyu Yang**[*1]    **Yunze Man**[*2]    **Jun-Kun Chen**[2]    **Yu-Xiong Wang**[2]

[1] Shanghai Jiao Tong University    [2] University of Illinois Urbana-Champaign

https://orangesodahub.github.io/SceneCraft

## Abstract

The creation of complex 3D scenes tailored to user specifications has been a tedious and challenging task with traditional 3D modeling tools. Although some pioneering methods have achieved automatic text-to-3D generation, they are generally limited to small-scale scenes with restricted control over the shape and texture. We introduce SceneCraft, a novel method for generating detailed indoor scenes that adhere to textual descriptions and spatial layout preferences provided by users. Central to our method is a rendering-based technique, which converts 3D semantic layouts into multi-view 2D proxy maps. Furthermore, we design a semantic and depth conditioned diffusion model to generate multi-view images, which are used to learn a neural radiance field (NeRF) as the final scene representation. Without the constraints of panorama image generation, we surpass previous methods in supporting complicated indoor space generation beyond a single room, even as complicated as a whole multi-bedroom apartment with irregular shapes and layouts. Through experimental analysis, we demonstrate that our method significantly outperforms existing approaches in complex indoor scene generation with diverse textures, consistent geometry, and realistic visual quality.

## 1 Introduction

The generation of diverse and complex 3D scenes plays a critical role in enhancing virtual and augmented reality (VR/AR) experiences, video game development, and the advancement of human-centric embodied AI. However, manually creating these complex 3D scenes is a tedious procedure that requires extensive knowledge and proficiency in 3D modeling tools [13, 14]. The recent success of 2D generative models [23, 50, 55] fuels the development of a line of text-to-3D work [31, 46, 63, 66]. Although these methods have achieved impressive object generation performance, scaling from object-level to scene-level generation presents significant challenges. It involves managing a considerably larger space with complicated semantics while ensuring 3D consistency (in terms of shape, texture, occlusion, *etc*.) across various camera perspectives.

Recent advances in scene-level 3D generation [17, 24, 37, 60, 75] have opened new pathways for creating larger-scale virtual environments. Most work leverages image inpainting [17, 24, 75] or multi-view diffusion methods [37, 60] to optimize a text-guided 3D scene. While generating locally convincing textured meshes, these methods share two common drawbacks: (1) Focusing on local coherence, they often struggle to accurately depict geometrically consistent rooms with plausible layouts and rich semantic details. (2) Conditioned only on textual prompts, these methods fall short in terms of offering precise control over the entire scene's composition and arrangement. Although some concurrent research [16, 45, 53] has explored the generation of an indoor environment conditioned on user-defined 3D layouts, it is restricted to creating small-scale compositions involving multiple

---

[*]Equal contribution.

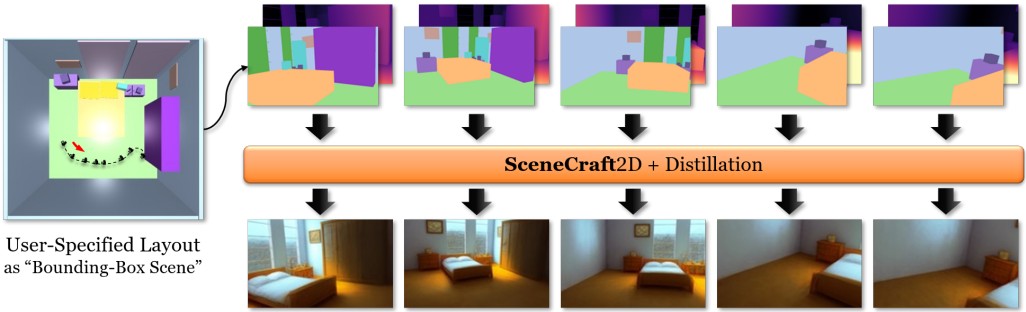

Figure 1: Our novel method generates complex and detailed indoor scenes from 3D spatial layouts and textual descriptions. Given user-specified layouts represented as a "Bounding Box Scene (BBS)," our method renders batches of 2D layouts and coarse depth maps and then transforms them into high-quality 3D scenes.

objects [12, 45], or lacks the ability to generate multiple rooms with complex layouts, shapes, and free camera viewpoints [16, 53] due to the use of panoramic representation.

In this paper, we introduce SceneCraft, a novel method designed to generate high-quality indoor scenes conditioned on user-specified *free-form* layouts. A high-level illustration of our work is shown in Figure 1. Our method features two key innovative designs:

**User-Friendly Semantic-Aware Layout Control.** Central to our approach is the utilization of 3D bounding boxes to guide the layouts of the target space, namely a "bounding-box scene (BBS)," which allows users to design complex and free-form room arrangements with simple bounding boxes. With this layout format, users can easily define both the spatial arrangement and the placement of objects within a room, as constructing a building in the Minecraft game. And SceneCraft leverages this preliminary design to generate a detailed and realistic scene. We surpass previous methods in supporting complicated indoor layouts beyond a single room, *even as complicated as a whole three-story house with multiple layers and irregular rooms.*

**High-Quality Complex Scene Generation with a 2D Diffusion Model.** Our framework excels in creating 3D scenes by leveraging the advanced generation capabilities of our pre-trained 2D diffusion model, SceneCraft2D. SceneCraft2D takes the "bounding-box images (BBI)" rendered from BBS as a condition through ControlNets [76] to generate high-fidelity views of the room that follow the given simple prompt like *"This is one view of a [style description] room."* By obtaining high-quality multi-view images through SceneCraft2D, we successfully distill a high-resolution 3D representation [56] of the generated indoor scene.

Trained with multi-view indoor scene datasets [49, 72], our work achieves state-of-the-art 3D indoor scene generation performance, both quantitatively and qualitatively. We present the *first* effective framework to generate complex text- and layout-guided 3D-consistent scenes with *free camera trajectories and diverse semantics.* In summary, our technical contributions are threefold:

- We propose a novel layout-guided 3D scene generation framework to create complicated indoor scenes adhering to user specifications, being the first to operate on free multi-view trajectories and free from the constraints of using panoramas.
- We introduce the "bounding-box scene" as a user-friendly format to scratch a desired room as easy as building homes in the Minecraft game, which provides accurate geometry control.
- We design a high-quality 2D diffusion model, SceneCraft2D, to generate high-fidelity and high-quality rooms following the rendered "bounding-box image" from the "bounding-box scene," and to support the generation of various styles via text conditioning.

With all these contributions, our SceneCraft achieves high-quality generation of various fine-grained and complicated indoor scenes that have not been supported by previous work.

## 2 Related Work

**Learnable Scene Representation.** Traditional scene representations [1, 11, 22, 28, 40, 44, 54, 64] directly model the 3D geometry information of the scene and thus suffer from limited flexibility

or low rendering quality. The neural radiance field (NeRF) [41] pioneers a neural network-based scene representation, providing the ability to reconstruct a complete and precise 3D scene from only multi-view images and corresponding camera parameters. Follow-up variants of NeRF [2, 6, 7, 20, 56, 62, 65, 67, 69, 73, 74] aim either to improve the original framework in different aspects, *e.g.*, rendering quality and training efficiency, or to support additional tasks, *e.g.*, relighting ability and editing ability. Recently, 3D Gaussian Splatting [27] outperforms the NeRF-family representation with high rendering quality and efficiency. In our work, we use a learnable scene representation as the backbone to model the output. As any representation can be used in our framework, we choose Nerfacto [56] for its high-quality rendering of complicated large-scale scenes.

**Diffusion-Guided Text-to-3D Generation.** The recent successful 2D diffusion-based generative models [3, 4, 23, 50, 51, 55] have inspired a series of innovative text-to-3D methods [10, 31, 43, 46, 59, 66, 75] to distill powerful 2D pre-trained models for 3D content creation. DreamFusion [46] proposes the score distillation sampling (SDS) module to optimize scene representations, *e.g.*, NeRF [41] or Gaussian Splatting [27], of objects by denoising their rendered views. Building on top of DreamFusion, SJC [63], Magic3D [31], and ProlificDreamer [66] alleviate the over-saturation problem and improve the generation quality. Despite impressive results, these methods are restricted in generating small-scale objects without complex semantic composition. More recently, Text2Room [24], SceneScape [17], and Text2NeRF [75] propose to extend object generation to scene generation with off-the-shelf text-image inpainting models [39, 50], where they iteratively inpaint unseen parts of the scene from novel camera perspectives. Another work [37] proposes progressively distilling a text-conditioned indoor panorama generation model [60] using different groups of camera views to optimize a scene representation. However, all of these methods lack semantic control over the generation output other than a simple text prompt. Hence, they cannot be used in the creation of 3D scene models where users want to specify the structure and layouts of the environment. In comparison, our work learns to generate 3D scenes that adhere to user-specified room layouts and textual descriptions, allowing precise control over the environment.

**Scene Generation with Semantic Guidance.** A recent line of work has studied 2D generation with semantic guidance for better controllability [9, 15, 18, 30, 50, 71]. Attempting to extend image generation to 3D creation, prior work has studied single image to 3D object reconstruction [33, 34, 36, 38, 47, 57, 68] and single image to video reconstruction [5, 61]. These methods face great challenges in 3D consistency, due to the lack of large scene-level datasets for training and the use of the auto-regressive generation paradigm. Meanwhile, Set-the-scene [12], CompoNeRF [32], and Compo3D [45] learn to generate object compositions from semantic layouts with the SDS method [46]. Discoscene [70] aims to disentangle the scene and then perform object-level scene editing leveraging the layout priors. DiffuScene [58] and GraphDreamer [19] utilize scene graphs together with textual descriptions as conditions to generate compositional 3D scenes. However, these methods are restricted to generating small-scale scenes composed of only several objects. They also neglect representations of walls, doors, ceilings, and ground, which are essential in defining indoor scenes but difficult to control in generation. Close to our work are three concurrent methods, ControlRoom3D [53], Ctrl-Room [16], and UrbanArchitect [35]. The first two methods generate 3D room meshes from user-defined or estimated layouts with multi-view diffusion followed by a monocular depth estimation process. While achieving outstanding generation performance, they rely on panorama images [60] as their preliminary results, which not only simplifies the scene generation problem, but also limits the complexity of their room layouts and diversity of their camera viewpoints. Our method, by learning a 3D-consistent multi-view generator and a scene renderer without viewpoint constraints, is able to generate more complex and consistent scenes with diverse camera trajectories. UrbanArchitect [35] focuses on street-view scene generation with semantic-aware layout controls. However, it allows for greater geometric approximation due to simpler conditions: fewer object categories, sparser and non-overlapping object placement, and more predictable camera trajectories. In contrast, indoor scenes feature dense objects that overlap with more fine-grained categories, which are all effectively addressed in our method.

## 3   SceneCraft: Methodology

Our SceneCraft is a novel method for text- and layout-guided scene generation. As illustrated in Figure 2, the input to SceneCraft consists of (1) a prompt as a coarse description of the target scene's

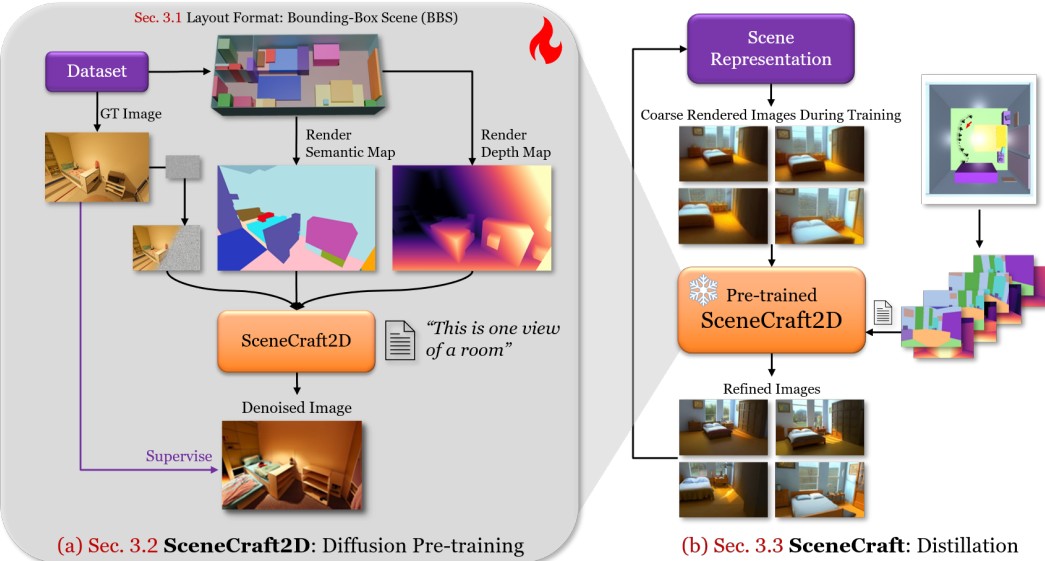

(a) Sec. 3.2 **SceneCraft2D**: Diffusion Pre-training     (b) Sec. 3.3 **SceneCraft**: Distillation

Figure 2: **SceneCraft** is a novel framework for layout-guided scene generation, which allows users to provide the layout as a bounding-box scene (BBS, Sec. 3.1), a user-friendly layout format that guides the generation. Our framework contains two stages: (a) pre-training of a 2D diffusion model, SceneCraft2D, to solve the 2D version of the layout-guided scene generation task (Sec. 3.2), and (b) distillation of the SceneCraft2D to learn a scene representation of the generated scene (Sec. 3.3).

style and content, (2) a "bounding-box scene" (BBS) serving as the layout guidance of the target scene, and (3) a camera trajectory defined in the space of BBS. SceneCraft renders the BBS in the camera trajectory to construct "bounding-box images" (BBI) as the layout condition for a pre-trained 2D diffusion model "SceneCraft2D" to generate high-quality 2D images of the scene. With the high-quality images generated by SceneCraft2D, SceneCraft is able to use an SDS-equivalent paradigm [46] to aggregate them into a scene representation (*e.g.*, NeRF [41] or 3D Gaussian splatting [27]) of the generated 3D scene. Notably, *our SceneCraft does not require a panoramic view*. Instead, our camera view can move freely in the 3D space, enabling the generation of much more complicated indoor layouts consisting of multiple rooms, unlike prior work which only supports single-room scenes.

## 3.1 Bounding-Box Scene (BBS): A User-Friendly Layout Interface

To provide a user-friendly format for free-form indoor layouts, we design the bounding-box scene (BBS) representation. As shown in Figure 2, BBS is similar to the "Proxy Room" of ControlRoom3D [53], but each object in the scene can be represented by a union of several intersecting bounding boxes in BBS with a category label, to indicate the coarse shape and category of an object. This provides users with the ability to indicate the shape of the object, *e.g.*, an L-shaped or even an S-shaped desk, while still maintaining the freedom of using a single bounding box for generation.

## 3.2 SceneCraft2D: Layout-Guided Image Generation

BBS can be regarded as a draft or a coarse version of the scene. In order to generate the actual room accurately conditioned on BBS, we use a distillation-guided framework. Each view of the generated scene corresponds to a 2D generation task, conditioned on the "bounding-box image (BBI)" of the same view in BBS, where each pixel of BBI contains both the semantic category and the depth of the pixel in BBS. By rendering BBS into BBI on the projected camera trajectory provided, we decompose the layout-guided 3D scene generation task into *a set of layout-guided 2D image generation tasks*, with BBI as conditions. To solve these tasks, we propose SceneCraft2D, a 2D diffusion model for high-quality layout-guided 2D image generation.

**Augmented SD for BBI Conditions.** Our SceneCraft2D is augmented from Stable Diffusion [50], with an additional BBI condition at the current viewpoint, which contains both the semantic category

map and the BBS depth map. The semantic map (converted to one-hot vectors based on the category) and depth map are injected into the model, as conditions via two separate ControlNets [76].

**Finetuning.** We finetune the augmented Stable Diffusion with scenes in indoor datasets like ScanNet++ [72] and Hypersim [49]. Each scene is converted to a generation task by generating the prompt, converting its semantic point cloud into a BBS, and using the camera trajectory provided by the dataset. We split the generation task into several 2D generation tasks at each view in the dataset, and train the SD model with these tasks. During the finetuning process, instead of using existing caption tools such as BLIP [29] to generate prompts, we use a single base prompt for all training samples. During inference-time generation, our model supports more specific and customized scene-specific prompts to produce the results that users desire. Note that the base prompt does not need to contain any information describing the image content, it merely serves as a placeholder to avoid the model overfitting to any particular word or sentence. Specifically, we use *"This is one view of a room."* as the base prompt and user-desired target prompts like *"This is one view of a bedroom in Van Gogh painting style."* for generation. The results showed that this method effectively controls the style of the generated outputs via prompts while maintaining a good layout-conditioned generation. After finetuning, SceneCraft2D can generate high-quality images according to the given BBI and text prompt.

### 3.3 Distillation-Guided Scene Generation

**Distillation Process with Annealing.** To generate 3D scenes, we distill the generation ability of our pre-trained SceneScraft2D model in a Score Distillation Sampling (SDS) [46, 63]-equivalent pipeline. Unlike the vanilla SDS [46] that works in the latent space and directly works with gradients, our pipeline applies an IN2N [21]-style, which is proven SDS-equivalent by HiFA [78]. In this pipeline, we maintain a multi-view dataset for continual scene representation training while simultaneously and iteratively replacing the multi-view dataset with newly generated images by SceneCraft2D. Through this process, the multi-view dataset will be gradually replaced with views of the generated scenes, which are used to fit the scene representation towards the generation.

Within this pipeline, we also propose an annealing-based distillation strategy inspired by [39, 78], for a more efficient and high-quality distillation. Leveraging the SDEdit method [39] to control the similarity of generated images with the currently modeled scene, we gradually decrease this similarity along with the entire distillation procedure. In other words, at an early stage of distillation, SceneCraft2D can freely generate the room to satisfy the BBS and the prompt; while at a later stage, by generating similar but higher-quality images, SceneCraft2D can also serve as a refiner of the scene representation to refine the rendering result and improve the scene representation. With this pipeline, our SceneCraft is able to generate high-quality scenes.

**Layout-Aware Depth Constraint.** When generating a complex indoor scene based on free camera trajectories, learning a reasonable geometry of the scene from scratch is both crucial and challenging. However, we have prior knowledge of the BBS input, which allows the model to quickly capture the geometry of the scene through the layout-aware depth constraint. Specifically, at the initial stage of distillation, we add a normalized depth loss $\mathcal{L}_{\text{depth}}$, where the pseudo-supervision signal comes from our BBS input. We set a soft threshold $\delta$ that allows the pixel depths $D_{\text{render}}$ modeled by the scene representation to fluctuate within a reasonable range around the pseudo-ground truth depths $D_{\text{layout}}$. This ensures that the model quickly converges to an initial coarse geometry. This loss is modeled in the following form:

$$\mathcal{L}_{\text{depth}} = [\max(||D_{\text{render}} - D_{\text{layout}}|| - \delta,\ 0)]^2. \tag{1}$$

Later in the distillation process, we disable this loss term to allow the model to learn more fine-grained geometry.

**Floc Removal with Periodical Migration.** The images generated at the initial steps of the distillation process have a lower consistency, which can result in blurry flocs close to the surface and in the air when "averaging" inconsistent multi-view images on the scene representation side. At a later stage, even when the diffusion's output is relatively 3D-consistent with annealing, the flocs, with condensed volume density, are still hard to remove and may result in Janus problems. Therefore, instead of "fixing" flocs issues in the original scene representation, we propose a method to migrate the current relatively coarse scene to another scene from scratch, to obtain a finer version. After the

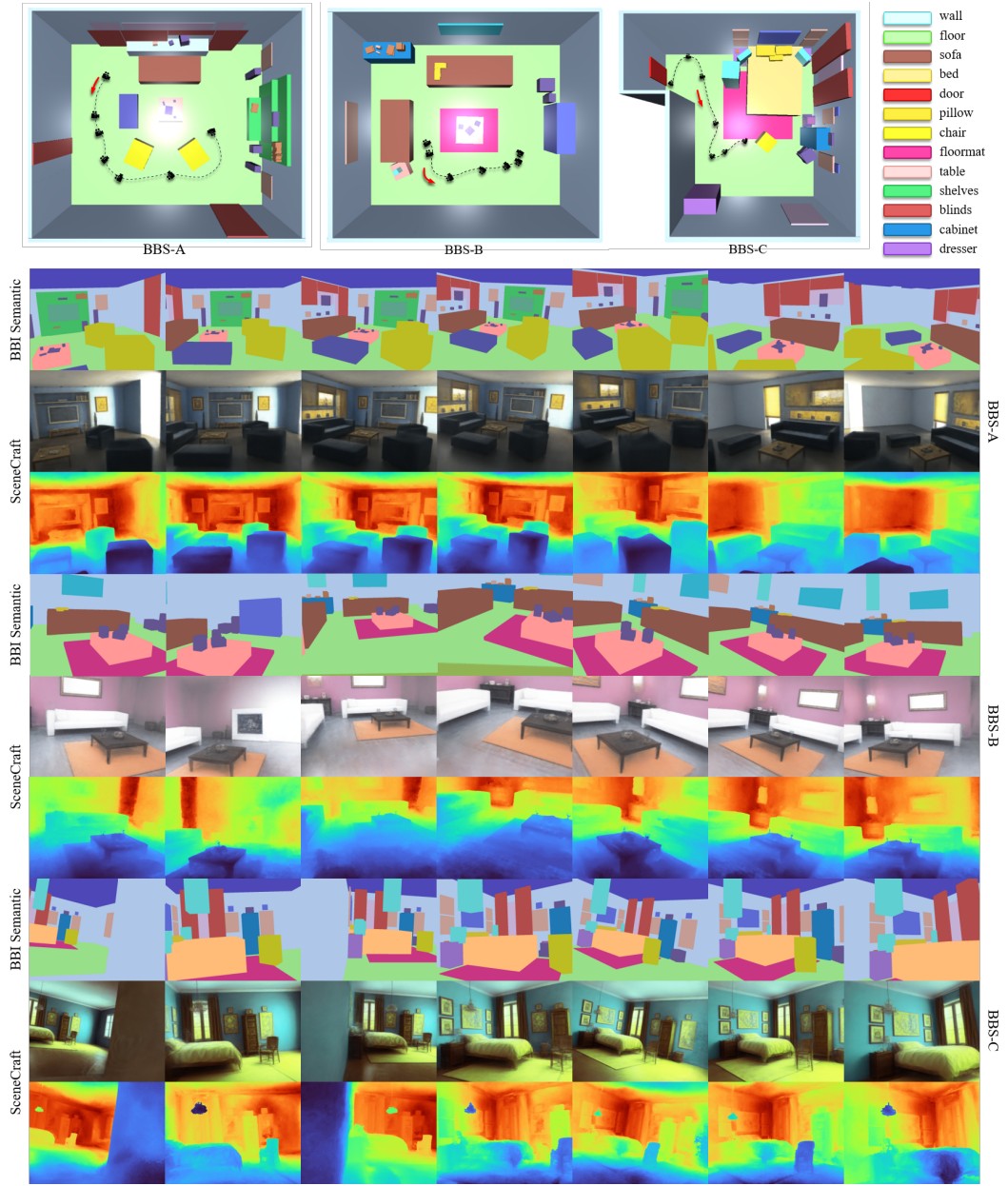

Figure 3: Generation results of **SceneCraft** on Hypersim [49] provided room layouts. For each sample, we demonstrate the 3D BBS and BBI semantic maps and the generated scene RGB images and rendered depth map. Our method is able to generate complex and free-form scenes from challenging room layouts.

first several iterations as early-stage training, we begin to maintain two scene representations, $S_c$ and $S_f$, to indicate the previous coarse representation and the mitigated fine representation, respectively. We freeze $S_c$ and generate new images to supervise $S_f$ by generating images similar to $S_c$'s rendering results (by only applying $t < T$ noise adding steps), to refine $S_c$ and store into $S_f$ with the diffusion model's generation. We also periodically update $S_f$ with $S_c$ (with a smaller interval of training iterations) to synchronize the latest information in both two scene representations. With the periodical migration method, we achieve more and more fine-grained and clear scenes during the training procedure.

**Texture Consolidation.** The generation of high-quality images by our SceneCraft2D ensures that the scene representation can converge accurately to the intended scene geometry. This advancement negates the necessity for explicit mesh exportation from scene representation as commonly required in previous work. To assign the modeled scene with sharp and clear textures, we incorporate the use

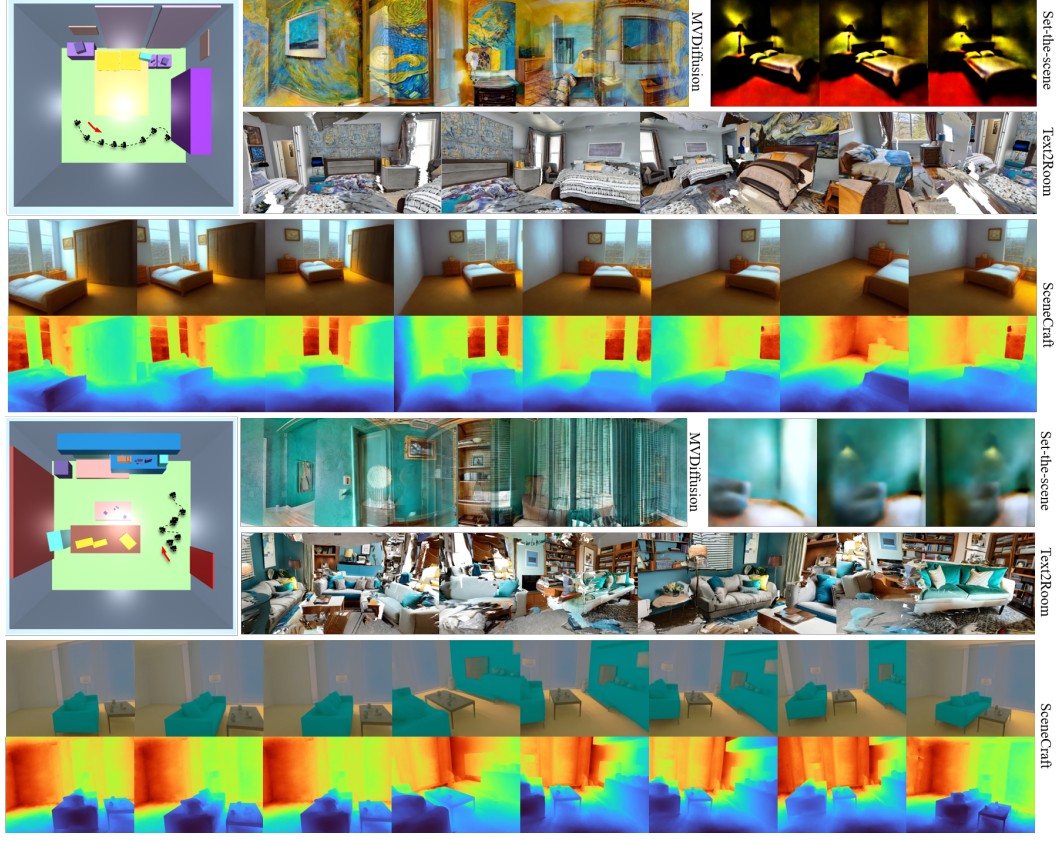

Figure 4: Qualitative comparisons of **SceneCraft** and baseline approaches. We show our generated color and depth renderings under two common layout conditions (a bedroom and a living room) alongside three other baselines. SceneCraft demonstrates higher credibility in following the layout conditions and is capable of handling more complex scenarios.

of VGG [25] perceptual and stylization loss during the distillation process. This strategy allows the scene representation to produce rendered images that share semantic meaning and stylistic elements with SceneCraft2D-generated images, rather than striving for pixel-perfect replication, which often leads to blurred results. By employing this loss, our SceneCraft framework emerges as a unified model to generate scenes in a sharp and clear manner, thereby eliminating the need for labor-intensive processes of mesh exportation and optimization.

## 4 Experiments

In this section, we focus on demonstrating the quality of SceneCraft generation under various layout conditions and prompts, and compare our performance with publicly available methods quantitatively and qualitatively. Then we present *more challenging generation that is beyond the scope of the previous methods.*

**Implementation and Datasets.** For the development of our SceneCraft2D diffusion model, we finetune Stable Diffusion [50] with our produced layout data. We use multi-view images from ScanNet++ [72] and HyperSim [49] to construct BBI data. In the distillation process, we choose Nerfacto from NeRFStudio [56] as our backbone for scene representation. During distillation, we use a dual-GPU pipeline to parallelize diffusion generation and NeRF training. More details are provided in Appendix Sec. A.

**BBS Sources.** For efficiency and effectiveness, we employ two distinct approaches to leverage bounding-box scenes (BBS), one of which utilizes original 3D bounding boxes (axis-aligned or oriented) by directly rendering them into 2D images. This straightforward method is already sufficient for the generation in our experiment, as we applied on Hypersim [49] data. Another approach enhances traditional bounding boxes by voxelizing them into a more detailed collection of smaller,

fine-grained voxels. This method is particularly adept at capturing the nuances of more complex geometries and arrangements within a scene, such as L-shaped tables or S-shaped desks, which often pose challenges for more simplistic modeling techniques. We find that this significantly improves the model's ability to accurately represent and understand the spatial dynamics and intricate designs of various objects within a scene. We use this strategy for realistic and challenging scenes [72].

## 4.1 Layout-Guided Scene Generation

**Baselines.** Most existing work does not generate scenes conditioned on user-specified layouts [17, 24, 37]. The only two concurrent scene generation methods that support layout guidance have not released their codebases [16, 53] for comparison. Hence, we make our best effort to create a fair comparison with open-sourced scene generation methods and demonstrate the effectiveness of our method through ablation study (Sec. 4.2). **Text2Room** [24] uses a text-conditioned inpainting model to construct the scene frame by frame. Following their original instructions, we change the text prompt along the trajectory to reflect which objects are visible in the current frame. Similarly, for **MVDiffusion** [60], we construct different prompts for each of the eight views that make up the panorama image. For **Set-the-scene** [12], we follow their official guidelines, using 3D modeling software (*e.g.*, Blender) to create the same layout input for training and set the same prompts as SceneCraft.

**Qualitative Results.** In Figure 3, we demonstrate qualitative generation results of SceneCraft on Hypersim [49] provided room layouts. These illustrations vividly demonstrate the model's proficiency in crafting detailed, complex, and free-form scenes, showcasing its application across both the realistic and synthetic datasets. Not only does it highlight the technical prowess of SceneCraft in navigating the intricacies of scene generation, but also its adaptability to the diverse requirements of real-world and artificially constructed environments. In Sec. 4.3, we demonstrate more challenging generation, which includes extremely challenging cases for panorama-based methods, but naturally supported by our framework.

**Quantitative Results.** We present quantitative comparisons with baseline methods [12, 24, 60] using both 2D and 3D metrics in Tab. 1. For 2D metrics, we compute the CLIP Score (CS) [48] and the Inception Score (IS) [52], which do not require ground truth scenes from the dataset, and therefore are agnostic to the dataset used in training. We also measure 3D quality by conducting a user study with 32 participants, who scored 3D consistency (3DC) and overall visual quality (VQ) of rooms generated by

Table 1: Quantitative comparisons of **SceneCraft** against baselines.

| Method | 2D Metrics | | 3D Quality | |
|---|---|---|---|---|
| | CS↑ | IS↑ | 3DC↑ | VQ↑ |
| Text2Room [24] | 22.98 | 4.20 | 3.11 | 3.06 |
| MVDiffusion [60] | 23.85 | **4.36** | 3.20 | 3.35 |
| Set-the-scene [12] | 21.32 | 2.98 | 3.53 | 2.41 |
| SceneCraft (Ours) | **24.34** | 3.54 | **3.71** | **3.56** |

different methods on a scale of 1 to 5. Our experimental design follows previous work [51]. The quantitative results highlight that our method consistently outperforms prior approaches in terms of the CLIP Score, 3D consistency, and visual quality. Regarding the Inception Score, we anticipate that our diffusion model's finetuning with fixed categories slightly limits generation diversity. However, this is not a major concern for our task, as previous work has struggled to achieve both high consistency and visual quality while being controlled by layout prompts. Additionally, we did not provide other common metrics on generative tasks, *e.g.*, Fréchet Inception Distance (FID) Score, since it is dependent on the ground truth dataset and would result in unfair and inaccurate comparison if applied to our experiments.

**Comparison with Existing Methods.** In Figure 4, we present our results compared with three baselines under two common layout conditions. SceneCraft significantly outperforms previous methods. For panorama-based methods (MVDiffusion), the biggest limitation lies in the inability to model rooms with complex shapes, such as L- or S-shaped structures. When using prompts to describe layout conditions, MVDiffusion fails to generate the desired results accurately. For inpainting-based methods (Text2Room), although they support free camera trajectories, their iterative

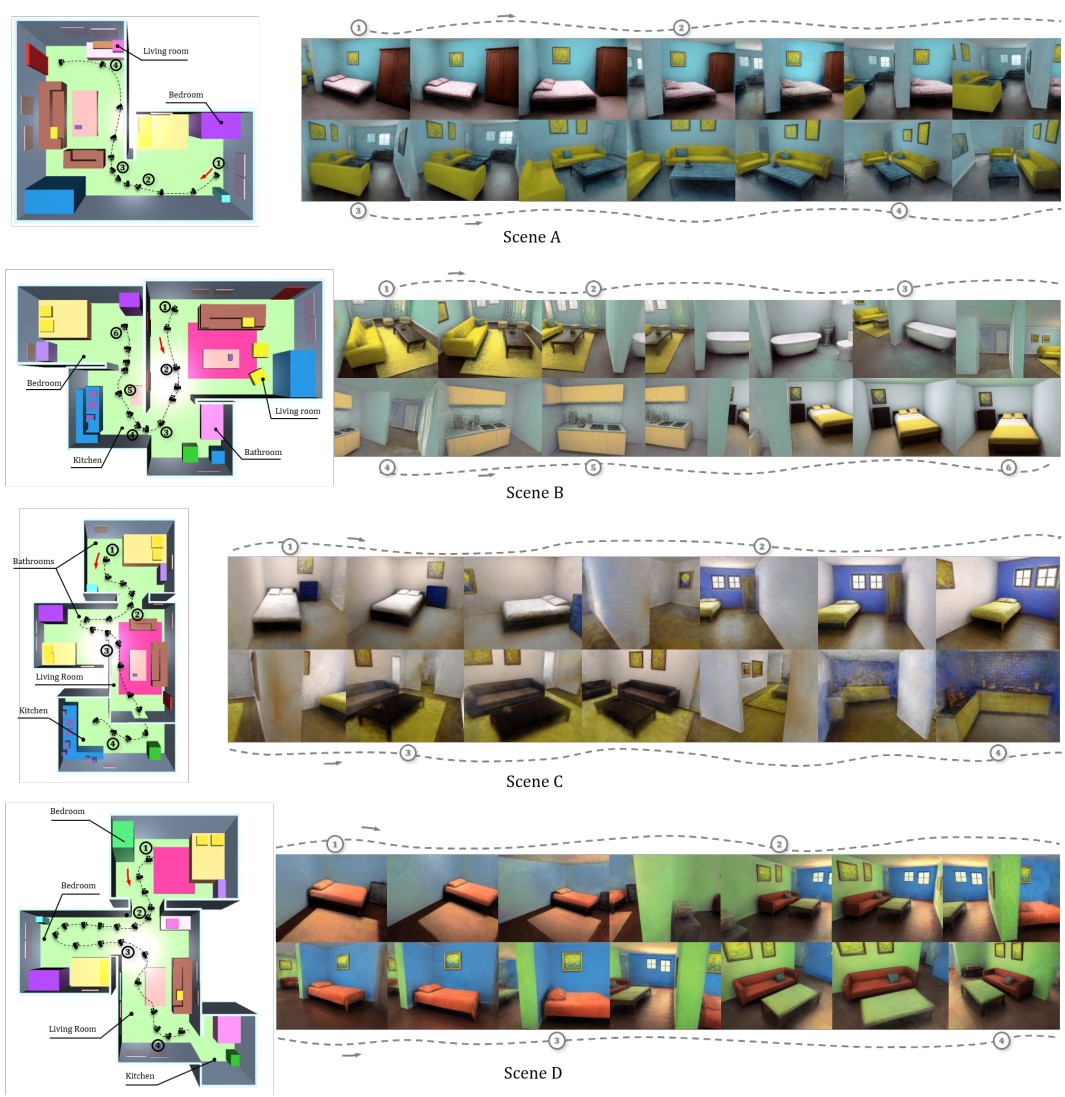

Figure 5: Generation results of **SceneCraft** in complex scenes. We demonstrate SceneCraft's ability to generate more complex indoor scenes leveraging arbitrary camera trajectories. Such non-regular shape of rooms cannot be naturally achieved by previous work.

generation nature often results in repetitive or contradictory frames. In the example shown in Figure 4, Text2Room generates four beds in that room simply because the prompt contains the word *"bedroom,"* completely failing to adhere to the specified layout conditions. For NeRF-composition methods (Set-the-scene), the main drawback is the inability to generate objects with significant size differences. Set-the-scene trains and combines different objects within the unified NeRF space. In Figure 4, Set-the-scene fails to generate objects hanging on walls, such as blinds or televisions. Our model, however, addresses all these issues: it can generate scenes of any scale and complexity following the given layout conditions and can also be adjusted via prompts.

## 4.2 Ablation Study

We conduct various ablation studies to validate our methods. Specifically, we test the effect of the base prompt used in finetuning, the layout-aware depth constraint, and the texture consolidation. The appendix section offers a comprehensive introduction to all of our evaluated models and additional experimental details, and includes further visualization and ablation experiments. We also elaborate on the limitations, failure cases, broader impacts, and future directions of our work. Please refer to Appendix Sec. B.1 for more details.

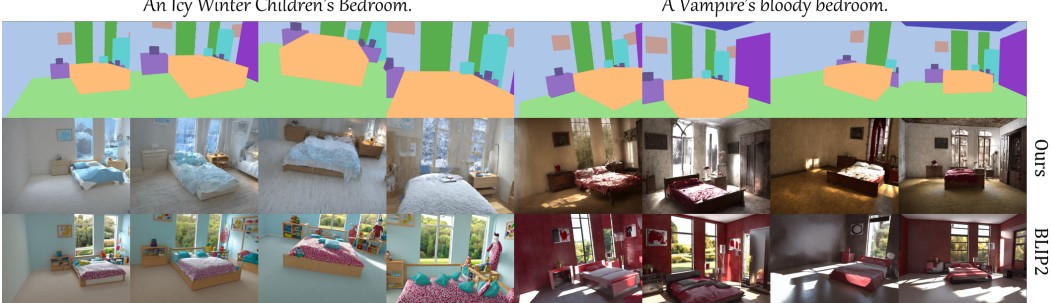

Figure 6: Effect of **Base Prompt**. Using our base prompt successfully avoids the overfitting and maintains the inherent power of pre-trained Stable Diffusion, while using BLIP2 captions leads to control failure.

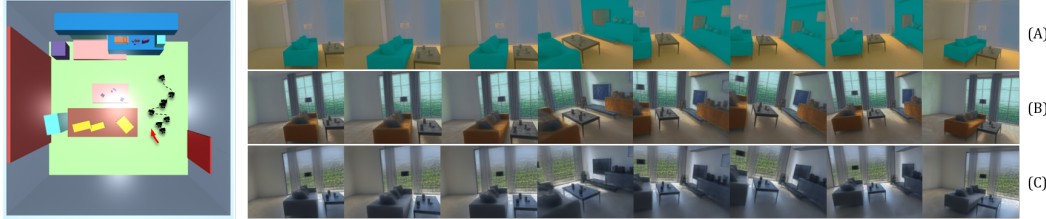

Figure 7: Style variants on the fixed layouts of **SceneCraft**. We show three variants A/B/C with different appearances while the geometries remain unchanged.

**Effect of Base Prompt.** To verify the effectiveness of using our base prompt, we test different prompt settings: *e.g.*, generating image captions from BLIP2 [29] with user-defined specific prompts. In Figure 6, we show that our method successfully achieves the control of the generation style through prompts, while maintaining a good layout-following ability. Considering the failure of the BLIP2 prompt and the complexity of our layout conditions, we believe that the more complex the condition, the more general the prompt we should take. In our case, we use "*This is one view of a room.*" to generally guide the model to fit the entire dataset of the indoor scene, rather than focusing on a particular class or object.

### 4.3 More Generation Results

**Generation on Irregular Shape.** In Figure 5, we showcase the results of more complex scene generation with fully customized layout on the free-camera trajectory. In the first example (Scene A), we customize an indoor layouts input where a bedroom is connected to a living room, along with the corresponding arbitrary camera trajectory. Theoretically, we can generate indoor scenes of any scale, for example, complex indoor room systems composed of multiple interconnected small rooms (Scenes B-D of Figure 5). Such tasks are not well-supported by methods based on panorama generation [60] or NeRF composition [12]. Although some other work [24] supports arbitrary camera trajectories, it performs poorly in establishing reasonable scene geometry and controlling the scene content.

**Style Variants Generation with Fixed Layouts.** In Figure 7, we show three variants of generation with the same room layout and different appearance, simply achieved by using different prompts. The results demonstrate the various control abilities of SceneCraft, allowing us to accurately define the shape and appearance of generation.

## 5 Conclusion

This work has introduced SceneCraft, an innovative method for generating complex and detailed indoor scenes from textual descriptions and spatial layouts. By leveraging a rendering-based operation, and a layout-conditioned diffusion model, our work effectively converts 3D semantic layouts into multi-view 2D images and learns a final scene representation that is not only consistent and realistic but also adheres closely to user specifications. Experimental results show the superiority of our model over existing state-of-the-art methods, highlighting its ability to generate diverse textures and maintain geometric consistency across complex indoor scenes.

## Acknowledgments

This work was supported in part by NSF Grant 2106825, NIFA Award 2020-67021-32799, the IBM-Illinois Discovery Accelerator Institute, the Toyota Research Institute, and the Jump ARCHES endowment through the Health Care Engineering Systems Center at Illinois and the OSF Foundation. This work used computational resources on NCSA Delta through allocations CIS220014 and CIS230012 from the Advanced Cyberinfrastructure Coordination Ecosystem: Services & Support (ACCESS) program, and on TACC Frontera through the National Artificial Intelligence Research Resource (NAIRR) Pilot.

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

# SceneCraft: Layout-Guided 3D Scene Generation

## Appendix

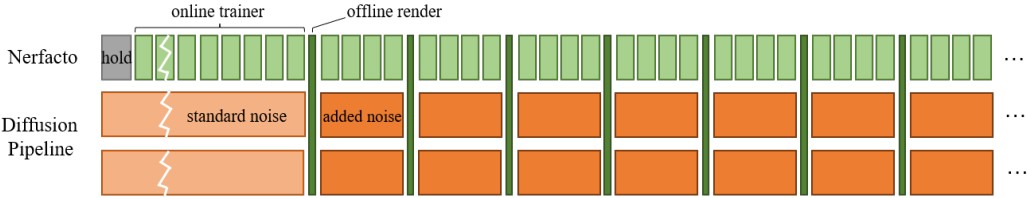

Figure A: An illustration of dual-GPU training scheduling.

## A  Additional Details

**Data Processing.**  As mentioned in Sec. 4, we use two datasets: HyperSim [49] and ScanNet++ [72], which are synthetic data and real-world data, respectively. We mainly use the HyperSim dataset in our experiments. For HyperSim, the original data provide approximately 77400 images of 461 indoor scenes, with the corresponding camera parameters and semantic bounding boxes. However, some of them do not meet the quality requirements of our training. Following prior work [53], we examine the entire dataset and filter out the lower-quality portions, such as rooms containing extremely complex and irregularly shaped objects, unbounded outdoor space, and rooms with excessively large scales (*e.g.*, churches, restaurants), ultimately retaining approximately half of the original data, amounting to around 24k pairs. ScanNet++ shows more complicated scenes (450+ scenes) compared to HyperSim. We voxelize them with a unit size equal to $0.2m$ to make the trade-off between rendering cost and data quality. Our processed data are publicly available [1] [2]. We leverage the Ray-OBB model (extended from the Ray-AABB model [26]) to complete the rasterization process.

**Dual-GPU Training Scheduling.**  Inspired by [8, 42], we have implemented a dual-GPU scheduler (see Figure A) for efficient diffusion and NeRF generation. Specifically, the second (or any GPU other than the first) GPU continuously generates new images to update the dataset, while the first GPU continuously trains Nerfacto with the current dataset. Once the diffusion procedure needs images from the first GPU to refine, the first GPU will switch to an offline renderer. This configuration effectively decouples the diffusion generation process, which is inherently more time-intensive, from the comparatively rapid NeRF training, thus streamlining the overall distillation workflow without compromising on quality or efficiency.

**Training Details and Cost.**  For finetuning the diffusion model, we use a total batch size of 16 on 2 NVIDIA A6000 GPUs with a constant learning rate of 5e-5, training for around 10k iterations. For the scene generation task, we use 2 A6000 GPUs to perform all our experiments. Generally, each well-generated scene takes around 3-4 hours with ~150 frames, 5-6 hours with ~300 frames (which handles camera trajectories of arbitrary length and complexity), the first GPU is responsible for the diffusion model and the memory cost is ~6GB in FP16 mode with an image size of 512×768. The other GPU is responsible for the scene representation process and the memory cost is ~28GB (with Nerfacto). Note that some concurrent methods cost more time: ShowRoom3D [37] costs approximately 10 hours to produce a single scene (Tab.3 in [37]), UrbanArchitect [35] costs ~12 hours and 32GB to produce a single scene (Sec. 4.1 in [35]). For NeRF training, we use a constant learning rate of 1e-2 for proposal networks and 1e-3 for fields.

## B  Additional Experiments

In this section, we first discuss additional aspects of ablation studies, which show the superiority of our method. Then we showcase additional generations on more complex layouts via our SceneCraft2D.

---

[1]Layout Scannet++: https://huggingface.co/datasets/gzzyyxy/layout_diffusion_scannetpp_voxel0.2
[2]Layout Hypersim: https://huggingface.co/datasets/gzzyyxy/layout_diffusion_hypersim

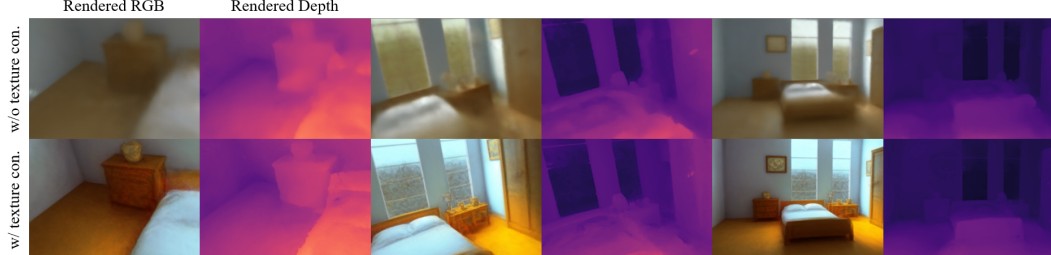

Figure C: Effect of **Texture Consolidation**. The generated renderings are blurry without the texture consolidation strategy. Our SceneCraft produces more detailed and textured results.

## B.1 Ablation Study

**Effect of Texture Consolidation.** We demonstrate the effectiveness of our texture consoidation shown in Figure C. Without texture consolidation, the loss function only includes the latent image loss and RGB image loss, similar to conventional SDS methods. In this case, the model fails to capture the high-frequency components of the scene from 2D images, resulting in very blurry generated scenes.

**Effect of Layout-Aware Depth Constraint.** We conduct experiments to validate the effectiveness of the layout-aware depth constraint. As depicted in Figure B, without this strategy, the model is completely unable to learn the correct geometry of the scene from 2D guidance. Although a reasonable appearance is achieved, it is mainly due to the fully flexible camera trajectories and complex layout conditions. Notably, using fixed camera poses as in panorama generation can partially

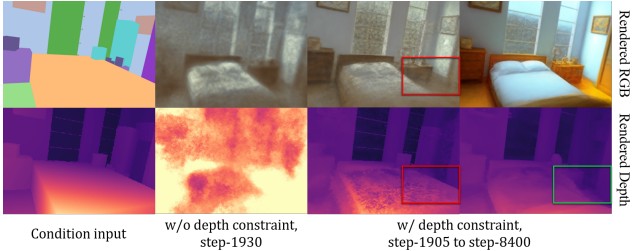

Figure B: **Layout-Aware Depth Constraint**. With depth constraint strategy, our SceneCraft can effectively learn scene geometry from prior input, which is crucial to ultimate 3D consistency.

alleviate this issue. With the layout-aware depth constraint, the scene geometry rapidly converges to the ground truth during the very initial training stage. Although some areas may find incorrect positions (red box), they are corrected during subsequent training after intermediately disabling this strategy, eventually leading to final convergence (green box). Once the geometry converges well, the color and texture begin to emerge.

## B.2 Complex Room Generation with Irregular Object Geometry and Free Camera Trajectory

In Figure D, we take the complex room layouts from the ScanNet++ [72] dataset as an example. The first two rows represent the input layout condition, and the last row shows the SceneCraft2D output. Note that for rooms with irregular object geometries, we convert voxelized 3D bounding boxes into more fine-grained voxels. This method particularly captures layouts with irregular geometries, such as L-shaped tables or S-shaped desks, which pose challenges to more simplistic modeling techniques. The results show that SceneCraft2D performs well in such complex room layouts.

## C  Limitations and Potential Future Direction

Although we have achieved promising results in complex scene generation from user prompts, 3D generation remains a very challenging task with many unsolved problems, and our method is limited in some aspects. This section provides a detailed discussion of the limitations and outlines potential future directions.

**Discussion on Failure Cases.** Despite our promising performance, some failure cases also exist: *(1) Extremely Complicated Scenes.* Our method may struggle to reason the layouts and generate rooms when layouts are excessively complex, containing many closely placed objects or highly overlapped bounding boxes (see Figure E). In this case, our optimized method using voxelization

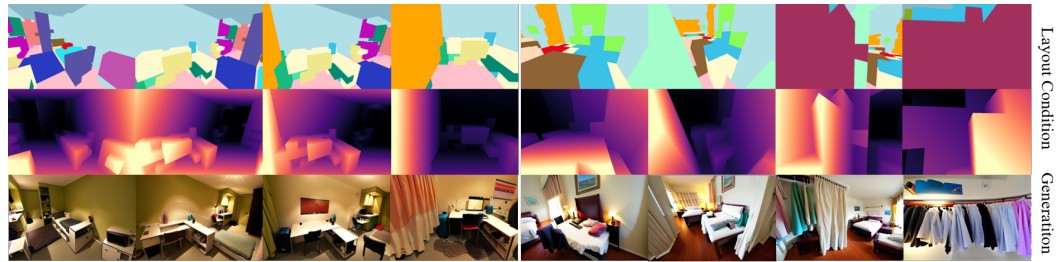

Figure D: Performance of **SceneCraft2D** on complex ScanNet++ provided room layouts.

does not provide clear and accurate representation of object layouts, which also reflects the difference between indoor and outdoor (street-view) scenes; *(2) Mismatched Layout and Prompt Inputs.* When the prompt does not align with the actual room layouts (*e.g.*, a bedroom layout with a "kitchen" prompt), our method may fail to generate appropriate room contents or achieve good convergence (see Figure F). Hence, users may need to adjust the corresponding prompts when generating a large complex scene with a long-term trajectory. All of these failure cases reflect the limitations of our method.

**Quality of Images Requires Further Improvement.** The quality of our generated 3D scenes still requires further improvement. When dealing with more challenging objects that typically have irregular geometries, such as hollowed-out chairs, lamps, or blinds, our results tend to be somewhat blurry. Moreover, complex layout condition inputs still limit the control ability of the prompt. For instance, we are unable to generate objects as vivid and richly detailed as those produced by the original diffusion model.

**Extension to Generation of Outdoor Scenes.** Most existing literature separates the task of generating indoor and outdoor scenes and focuses on one of the scenarios in their work. The generation of indoor and outdoor scenarios features different challenges, with indoor scenes having denser and more complex layouts, and outdoor scenes having more dynamic objects and a larger space to cover. Although it is a valid choice to start from indoor scene generation because of its relatively smaller scale and complex layouts, it is not comprehensive. The generation of outdoor scenes will likely lead to unique challenges and insights, and we consider this a direct future direction.

**Other Future Directions.** There are also challenges in defining fair, accurate, and comprehensive metrics for evaluating 3D scene generation methods in all aspects. The development of such metrics remains an open challenge and serves as a potential direction for exploration in the generative AI community. Another promising direction is flexible and controllable scene editing using decomposed 3D representations, given that our layout input is freely defined and adjustable. For complex scene generation, creating layouts by hand could be time-consuming and laborious, hence integrating some methods of automatic scene layouts and camera trajectories creation will also be a future topic, such as LLM (large language model)-based [77] and transformer-based approaches [16] . Extending our framework to incorporate user feedback loops for iterative refinement of generated scenes offers another promising direction.

# D   Societal Impact

We anticipate a potential positive social impact from our work. Being able to generate high-quality 3D digital content based on user queries, SceneCraft has the potential to revolutionize various industries, from VR/AR to architectural design and gaming. By significantly reducing the time and expertise required to create detailed 3D scenes, our work also leads to more accessible and fair content creation.

**Potential Negative Societal Impact.** We do not see a direct negative societal impact of our work. Indirect potential negative impact involves misusing scene generation models for digital content creation. We believe that it is crucial for researchers to proactively consider these concerns and establish guidelines to ensure responsible usage of these models.

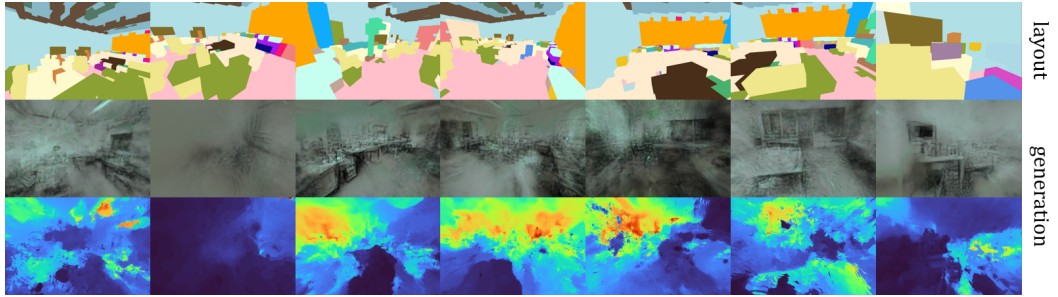

Figure E: Failure case #1: Extremely complicated scene that contains many closely placed objects and even highly overlapped bounding boxes.

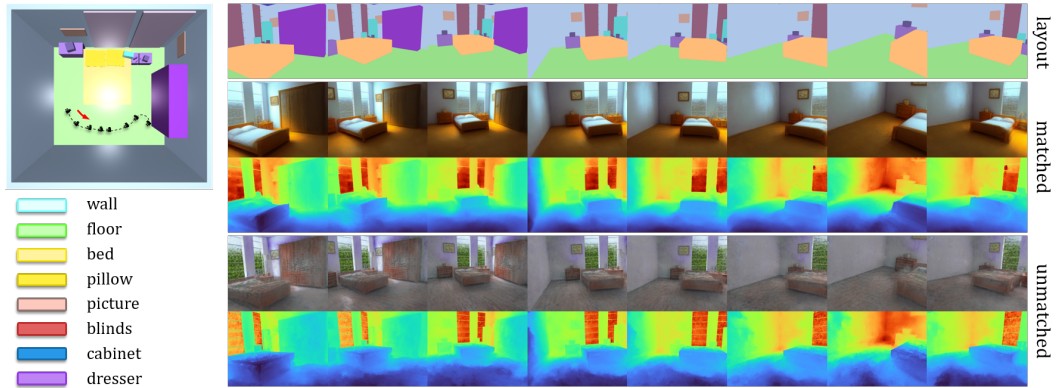

- wall
- floor
- bed
- pillow
- picture
- blinds
- cabinet
- dresser

Figure F: Failure case #2: Generations of a bedroom with a matched prompt "Bedroom" and a mismatched prompt "Kitchen."

# E    License of Dataset Used

In this section, we list the licenses of all the datasets we have used during our evaluation.

- ScanNet++ [72]: The ScanNet++ data are released under the ScanNet++ Terms of Use[3].
- HyperSim [49]: Creative Commons Attribution-ShareAlike 3.0 Unported License.

In addition, we utilize a number of public foundation model checkpoints pre-trained on various data sources in our paper. Please refer to their original papers [39, 50] for the license of datasets they have used in pre-training their models.

---

[3] https://kaldir.vc.in.tum.de/scannetpp/static/scannetpp-terms-of-use.pdf

