# OpenReview forum: "SceneCraft: Layout-Guided 3D Scene Generation"
_NeurIPS.cc/2024/Conference — NeurIPS 2024 poster_

### Official Review · Reviewer_DHDm · 2024-07-10

**Soundness:** 3
**Presentation:** 3
**Contribution:** 2
**Rating:** 5
**Confidence:** 4

**Summary:**

This paper proposes a layout-guided method for room generation. The proposed method uses 3D scene layout as a prior and leverages semantic and depth maps as 2D conditions. Specifically, the authors start by training a 2D conditional diffusion model named DreamScene2D, which utilizes ControlNet to incorporate projected semantic and depth maps and conditions. Then the 3D scene can be distillated from the conditional model. The proposed method achieves good rendering quality in complex room scenes.

**Strengths:**

1. Using 3D layout as a prior for 3D scene generation is reasonable, which is a user-friendly format for scene control.
2. The proposed method achieves high rendering quality and good controllability. The generated results can be controlled using both text prompts and scene layouts.
3. The overall writing is easy to follow.

**Weaknesses:**

1. The key insight of the proposed method is similar to recently released paper UrbanArchitect [1], which also uses layout as prior to generate urban scenes (more complex and larger scale). DreamScene uses a pipeline based on InstructNerf2Nerf and UrbanArchitect directly uses a layout-guided SDS. Although the two works can be seen as concurrent works, I hope the authors can include a discussion.
2. As a 3D generation model, it would be better to provide video results for better illustration. The authors can also provide 3D results for better visualization (e.g., generated 3D mesh like text2room).
3. Quantitative results are missing. The authors should at least perform User Study regarding the visual quality and the 3D consistency. Besides, since the ControlNet is trained on specific datasets, metrics like FID can also be used for better evaluation.
4. Some training details are missing, e.g., requirements for memory and time.
5. Since the layout prior is used, it would be desired to use decomposed 3D representations for further editing.
6. It would be better to include a method for automatic scene layout creation. For example, Ctrl-Room utilizes a scene code diffusion mode to generate scene layouts [2]. LLM can also be used to generate scene layouts [3].

[1] Lu, Fan, et al. "Urban Architect: Steerable 3D Urban Scene Generation with Layout Prior." arXiv preprint arXiv:2404.06780 (2024).

[2] Fang, Chuan, et al. "Ctrl-room: Controllable text-to-3d room meshes generation with layout constraints." arXiv preprint arXiv:2310.03602 (2023).

[3] Hu, Ziniu, et al. "SceneCraft: An LLM Agent for Synthesizing 3D Scenes as Blender Code." Forty-first International Conference on Machine Learning. 2024.

**Questions:**

1. 3DGS has been widely used in 3D generation models due to its high speed and good quality. Have you tried to use 3DGS to replace NeRF?
2. Is there a limit to the complexity of the scenarios a method can handle?
3. Text2Room uses SD for missing parts inpainting. Have you tried to replace SD model in Text2room with your finetuned SD model for inpainting for controllability?

**Limitations:**

Please refer to weaknesses.

---

> ### Author Rebuttal · Authors · 2024-08-07
>
> We appreciate your comments, and address your concerns as follows:
>
> ***
> 1. *Q: The key insight of the proposed method is similar to the recently released paper UrbanArchitect [A]. Although two works can be seen as concurrent works, I hope the authors can include a discussion.*
>
> We appreciate the reviewer highlighting this related work. We were unaware of UrbanArchitect [A], which was released just before the submission deadline. We will include a discussion of this concurrent work in our revision.
>
> We would like to clarify that outdoor scenes are not necessarily more complex than indoor scenes.
> * Indoor scenes feature dense, overlapping bounding boxes with fine-grained categories, while outdoor scenes typically have sparser, non-overlapping boxes with broader categories. Indoor scene generation methods must therefore handle object overlap and reason about relationships among numerous objects of varied categories.
> * Outdoor scene generation allows for greater geometric approximation. As evidenced in Fig.17 of [A], while street views are reasonably generated, peripheral elements like houses and cars lack precise construction. Indoor scenes, conversely, demand accurate geometry for all objects within a closed room.
> * Our distillation pipeline is optimized for efficiency and reduced memory usage. In contrast, [A] employs VSD and a two-stage generation process, incurring higher time and memory costs.
> * As noted by HiFA [71] and mentioned at L124, the SDS used by [A] is essentially equivalent to the IN2N we employ in DreamScene, with only slight differences in loss calculation. Building on IN2N, we introduce a novel distillation pipeline that removes artifacts through periodic migration and enhances texture consistency via specialized loss design (Sec. 3.3, L165-184), achieving superior 3D consistency.
>
> In conclusion, indoor and outdoor scene generation present distinct challenges and cannot be directly compared. Methods developed for outdoor scenes may not be directly applicable to indoor scene generation tasks.
>
> ***
> 2. *Q: Visualization: it would be better to provide video results for better illustration.*
>
> Thank the reviewer for the suggestion. Please refer to the general response for our provided video visualization.
>
>  ***
> 3. *Q: Evaluation: authors should provide quantitative results.*
>
> We appreciate the reviewer's suggestion. We have provided the results in the general response.
>
> ***
> 4. *Q: Training details: Some training details are missing (requirements for memory and time)*
>
> We thank the reviewer for the suggestion. We have provided the training details in the general response.
>
> ***
> 5. *Q: Since the layout prior is used, would it be desired to use decomposed 3D representations for further editing?*
>
> While layout-controlled scene editing is indeed an intriguing concept, we believe it falls outside the scope of our current manuscript. Our focus, in line with existing indoor scene generation methods [11,24,55], has been primarily on the generation task. Given the time constraints of the rebuttal phase, we have not explored this avenue thoroughly. However, we acknowledge the potential of this idea and consider it a promising direction for future research.
>
> ***
> 6. *Q: It would be better to include a method for automatic scene layout creation. E.g., Ctrl-Room utilizes a diffusion model [A]. LLM can also be used [B].*
>
> We would like to clarify that our method is designed to be layout-agnostic, accepting scene layouts as input regardless of their source. This flexibility allows for layouts to be pre-defined, user-generated, or even LLM-generated. Hence, our framework is compatible with various layout generation methods, including LLM-based approaches like Ctrl-Room [16]. This means that a text-guided layout generation module could be seamlessly integrated into our system to support fully text-guided scene generation.
>
> Given the time constraints of the rebuttal period, we have not implemented this extension. However, we recognize its potential and plan to include it in our final revision. This addition would further enhance the versatility and user-friendliness of our system without altering its core functionality.
>
> ***
> 7. *Q: 3DGS has been widely used in 3D generation models. Have you tried to use 3DGS to replace NeRF?*
>
> As mentioned in our introduction, our method is designed to be agnostic to the underlying scene representation, allowing for compatibility with various approaches. Our choice of NeRF was based on its proven stability in handling large-scale indoor scenes. However, we concur that integrating 3DGS could potentially offer comparable performance with improved convergence time. Due to the limited time in the rebuttal phase, we will try 3DGS in our revision.
>
> ***
> 8. *Q: Is there a limit to the complexity of the scenarios a method can handle?*
>
> Thank the reviewer for considering this problem. We provide a discussion around failure cases and in the **general response**.
>
> ***
> 9. *Q: Text2Room uses SD for missing parts inpainting. Have you tried to replace the SD model in Text2room with your finetuned SD model for inpainting for controllability?*
>
> We appreciate the reviewer's insightful suggestion.
> * Indeed, our diffusion model could potentially replace SD in Text2Room for inpainting tasks, as it is specifically designed for indoor scene generation. Moreover, this substitution could introduce layout conditioning capabilities to the Text2Room framework, potentially enhancing its controllability.
> * However, it's important to note that Text2Room relies on a depth-based naive 3D reconstruction approach. This fundamental aspect of its architecture would likely still result in challenges related to inappropriate or incomplete geometry, regardless of the diffusion model used for inpainting. Consequently, we anticipate that the results may not achieve the same level of quality as our current method.
> * Due to the time limitation, we consider this investigation as future work.

---

> > ### Comment · Area_Chair_bT2w · 2024-08-11
> >
> > Dear Reviewer,
> >
> > This is a gentle reminder to please review the rebuttal provided by the authors. Your feedback is crucial to the decision-making process. Please consider updating your score after reading the rebuttal.
> >
> > Thank you for your help with the NeurIPS!
> >
> > Best, Your AC

---

> > ### Comment · Reviewer_DHDm · 2024-08-13
> > **Response to rebuttal**
> >
> > The authors' rebuttal has addressed some of my concerns (training details, quantitative results, video results, etc.). Several experimental results are still missing (comparison with SDS-based pipeline, layout-enhanced Text2Room). Considering the weaknesses discussed above, I keep my score at Borderline Accept.

---

> > > ### Author Response · Authors · 2024-08-14
> > >
> > > We appreciate the reviewer for checking our rebuttal.
> > >
> > > 1. **For SDS-based pipeline**, we believe that you are referring to the comparison with recent work UrbanArchitect.
> > >    1. We would like to point out that UrbanArchitect is an arXiv manuscript released just before the NeurIPS submission deadline, which, as the reviewer also noted, should be considered as a concurrent work of ours. Not comparing with this method should not be regarded as a weakness of ours, especially since this method focuses on a different task than ours (outdoor scene v.s. indoor scene).
> > >    2. Due to the limited time in the rebuttal phase, and the substantial difference between the outdoor and indoor scene generation settings (as we have discussed in Q1 of the rebuttal), we are unable to finish training of UrbanArchitect with any indoor dataset to use it for indoor scene generation.
> > >    3. As mentioned in the previous rebuttal, we will cite and discuss UrbanArchitect in the revision.
> > > 2. For **layout-enhanced Text2Room**:
> > >    1. We acknowledge that this can be seen as an interesting exploration. However, we believe this is weaker than the baselines used in our submission, as Text2Room relies on a depth-based naive 3D reconstruction approach.
> > >       1. This fundamental aspect of its architecture would likely still result in challenges related to inappropriate or incomplete geometry, regardless of the diffusion model used for inpainting.
> > >       2. Simply replacing the diffusion model of Text2Room to our DreamScene2D could only enable the control of the appearance of generated images, but cannot at all help with the geometry issues resulting from the naive reconstruction, as such reconstruction is still only based on a weak monocular depth estimator.
> > >       3. Hence, we anticipate that such a model is unlikely to achieve the same level of quality as our current method. Importantly, lacking this ablation study does not demolish the contribution of our work, which was already demonstrated by our state-of-the-art results.
> > >    2. For the reviewer’s reference, we would also like to point out that previous published work Ctrl-Room [16] and ControlRoom3D [48], while performing layout-guided scene generation, do not include this additional experiment.
> > >    3. Per the reviewer’s request, we have already implemented this baseline and the model is training. However, due to the limited time in the rebuttal phase, we are not able to get the final results before the end of the discussion phase. The current preliminary results indicate that this method is significantly worse than ours. We will add this experiment in the final revision.
> > >
> > > We hope that our clarifications and additional results will address the reviewer’s remaining concerns. We look forward to a potentially more favorable evaluation of our work.

---

### Official Review · Reviewer_Gzhh · 2024-07-12

**Soundness:** 3
**Presentation:** 2
**Contribution:** 2
**Rating:** 6
**Confidence:** 3

**Summary:**

This work tackles the problem of 3D scene generation from bounding box scene, text prompt and camera trajectory. The core contribution of the work stems from a 3D scene generation method where the method is able to produce a complex scene, which the previous works could not generate due to using a panoramic representation. The method generates 2D images from multiview images using a diffusion model and then generates 3D. The results show that method is able to generate realistic 3D scenes from the user input.

**Strengths:**

- The method makes sense. Conditioned on the semantic mask, depth map, and text from a viewpoint in the 3D scene, the work uses diffusion model to generate scenes and uses NeRF to reconstruct 3D scenes. To increase 3D consistency the authors use floc removal with periodical migration.
- Using the bbox layout seems a nice way to control the 3D scene generation, since the users usually have a specific layout in mind.
- The figures show a nice portrait of the method.

**Weaknesses:**

- The scenes look more synthetic compared to the baselines such as MVDiffusion and the text2room.
- More qualitative results are required to be convincing. What fascinates me about this work is that the method is able to produce a complex layout scene. However, there is only a single example in the appendix, which the panoramic scene methods cannot generate. The authors should provide more qualitative examples to show that the method can indeed generate complex scenes. Also, the authors claimed that the method can generate various scenes, but only few of the visual examples were provided.
- In line 55, the authors mentioned that the method achieves SOTA performance quantitatively. Could the authors provide the quantitative results to back the claim? Also, I would like to see what the “qualitative” here refers to.
- In Line 66, the authors mentioned “our DreamScene achieves high-quality generation in various fine-grained and complicated indoor scenes, which are never supported by previous work”. To me this sounds a little bit confusing. Previous works such as ControlRoom3D were able to generate various high-quality scenes, yet they were not able to generate complicated indoor scenes. I think the authors should clearly mention that the completed indoor scenes is the key difference between the previous works.

**Questions:**

- I’m not fully understanding why camera trajectory is required to generate a 3D scene. If the users no the bounding box, couldn’t the method just use random camera trajectory to make the method work?
- In line 134, the authors mentioned “.glet@tokeneonedot”, what does this mean?
- What is the approximate run time of the method? Since the method generates 2D images and generate 3D, it maybe slow.

**Limitations:**

The method is limited to generation of a small scene. It would be nice to provide results on a large outdoor scene, which does not anyway diminish the work's contribution.

---

> ### Author Rebuttal · Authors · 2024-08-07
>
> We appreciate your comments, and address your concerns as follows:
>
> ***
> 1. *Q: Overall, the scenes look more synthetic compared to MVDiffusion and text2room.*
>
> We'd like to address this concern of the reviewer by providing context for our approach:
> * **Prioritizing 3D Consistency**: Our primary focus in this work is achieving 3D consistency in generated rooms, which we consider the most crucial and fundamental factor in 3D generation tasks, especially for complex room layouts. While this approach may result in some loss of detail or texture simplification, it consistently produces rooms with reasonable geometry that accurately matches the specified layout.
> * **Differences in Scene Representation**:
>     * The two methods mentioned have different scene representations from ours. MVDiffusion and Text2Room use RGB images from diffusion models directly and reconstruct scenes using depth estimators.
>     * Our method employs Neural Radiance Fields (NeRFs), offering a different approach to scene representation.
> * **Dataset Choice and Impact**:
>     * Our diffusion model is primarily fine-tuned on the synthetic Hypersim [45] dataset, chosen for its high-quality layout data. This choice aligns with other recent works like ControlRoom3D [48].
>     * In contrast, MVDiffusion uses a version of Stable Diffusion fine-tuned on the real-world ScanNet dataset, while Text2Room uses the original Stable Diffusion model.
> * **Flexibility for Improvement**: The perceived quality of our results can be significantly enhanced by adjusting prompt instructions, distillation parameters, and camera trajectories. One example is provided in the **General Response**.
>
> In summary, while our current results may appear more synthetic in some aspects, this is not an inherent limitation of our method. Rather, it reflects our current focus on geometric consistency and layout fidelity. Our approach offers significant flexibility for quality improvements while maintaining its core strengths in 3D consistency for complex scenes.
>
> ***
> 2. *Q: In order to really demonstrate the uniqueness of this work, authors should provide more challenging qualitative results (especially complex scenes).*
>
> In our main paper, we aimed to show the diversity (i.e., different room types and styles) and layout-controllability of our model with the results. And considering the results shown in our paper, we believe those adequately demonstrate the superiority over the sota baselines [11,24,55].
>
> Following your suggestion, we provide more results on complex scene generation in the rebuttal PDF and the linked video, as detailed in the **General Response**.
>
> ***
> 3. *Q: Evaluation: authors should provide quantitative results.*
>
> We appreciate the reviewer's suggestion for quantitative results. We have provided the results in the general response.
>
> ***
> 4. *Q: “our DreamScene achieves … and complicated indoor scenes, …”. This sentence seems confusing. Authors should clearly mention that (being able to generate) complicated indoor scenes is the key difference to previous works.*
>
> We thank the reviewer for pointing this out. We will revise and clarify this in our revision.
>
> ***
> 5. *Q: Why couldn't we just use a random camera trajectory?*
>
> It is important to note that generating a reasonable camera trajectory, particularly for non-square rooms with complex layouts, is far from trivial. There are several key considerations:
> * **Complexity of Trajectory Selection**: In complex room layouts, a random camera trajectory may intersect with objects or fail to cover all scene elements adequately. Even in 2D scenarios, collision-free trajectory generation remains a challenging research problem in robotics. There are dedicated papers investigating this problem in the problem of text-to-3D generation [A, B], suggesting its complexity.
>     * *[A] E.T. the Exceptional Trajectories: Text-to-camera-trajectory generation with character awareness, ECCV 2024*
>     * *[B] CameraCtrl: Enabling Camera Control for Video Diffusion Models, arXiv 2024*
> * **Consistency with Prior Work**: Our approach aligns with existing methods in the field. For instance, Text2Room also requires user-provided camera trajectories. Other baselines that use panoramas have a built-in camera trajectory (rotation around a center point), but this limits them to simple square rooms with minimal occlusions.
> * **Advantages of current setting**: By allowing for specified camera trajectories, we enable more precise control over scene exploration and visualization, which is particularly valuable in complex room layouts. This approach provides flexibility for future integration with advanced trajectory generation methods as they develop.
>
> ***
> 6. *Q: Typo at line 134?*
>
> We will revise the manuscript and correct this typo.
>
> ***
> 7. *Q: Training details: Lack of runtime data?*
>
> We have provided the training details in the general response.
>
> ***
> 8. *Q: The method is limited to small scenes. It would be nice to provide results on a large outdoor scene, which does not anyway diminish the work's contribution.*
>
> We appreciate the reviewer's observation. Regarding the generation of large outdoor scenes:
> * This is not a limitation of our method. While large outdoor scenes might intuitively seem more challenging due to their scale, our work prioritizes 3D-consistent, complex layout-controlled generation — a challenging aspect independent of scene size.
> * Outdoor scenes generally present simpler conditions: fewer object categories, sparser object placement, and more predictable camera trajectories. For instance, most outdoor scene views follow a common paradigm (road and sky in the center, vehicles and buildings on the sides), simplifying the diffusion process.
> * Our focus on dense indoor scenes aligns with the problem setting of several prior methods [11,24,55]. We consider outdoor scene generation beyond the scope of our current model, but we acknowledge it as an intriguing direction for future research.

---

> > ### Comment · Area_Chair_bT2w · 2024-08-11
> >
> > Dear Reviewer,
> >
> > This is a gentle reminder to please review the rebuttal provided by the authors. Your feedback is crucial to the decision-making process. Please consider updating your score after reading the rebuttal.
> >
> > Thank you for your help with the NeurIPS!
> >
> > Best, Your AC

---

> > ### Comment · Reviewer_Gzhh · 2024-08-12
> > **Could the authors provide more qualititative results if possible?**
> >
> > Thank you for the fruitful response. I really appreciate the author's effort for the response.
> >
> > I know this is a last minute response (and I am sorry for this), but could the authors provide more qualitative results on complex scenes? I ask this because the main argument of the proposed method is that the method can generate complex scenes, however, the only complex scenes in the manuscript and the rebuttal is the 2 scenes in the rebuttal (the scene in the appendix has the same L shaped room as in the rebuttal, otherwise the scenes look like a bbox to me). It's not very convincing to believe that the proposed method can generate complex scenes with just 2 (complex scene) visualizations. I reallize that the authors have only one more day for the rebuttal, but could the authors provide more examples with complex scenes?

---

> ### Author Response · Authors · 2024-08-14
> **Reply to reviewer Gzhh’s feedback**
>
> We appreciate the reviewer's feedback on the need for more complex scene examples.
> - During the limited rebuttal period, we have tried our best to generate two new complex scenes, as shown in this anonymous link: https://github.com/DreamScene3D/DreamScene/blob/main/rebuttal-visualization.jpg?raw=true. The last two rows showcase these new results.
> - We would also like to point out that “complex scenes” are not just limited to non-square shaped ones – square scenes can also be complex so that the baselines will fail. In our Fig.3, the BBS-A/B/C contains many objects that are not adjacent to the walls and corners, which will also create occlusions in panoramas and make the depth map non-continuous and hard to predict. Therefore, these scenes should also be regarded as “complex scenes” where our DreamScene is the only work that can work well.
>
> We hope these results could establish our DreamScene’s ability to generate rooms based on various complicated layouts, and address your concern about this. We commit to including these new examples, along with additional complex scene generations, in our future revision.

---

### Official Review · Reviewer_Nerf · 2024-07-15

**Soundness:** 2
**Presentation:** 3
**Contribution:** 3
**Rating:** 6
**Confidence:** 3

**Summary:**

DreamScene proposes a method for 3D indoor scene generation with text and layout as input. For this, they propose using a 3D bounding box scene representation as the means to provide layout guidance. This is then used as an input, along with text prompt, to a 2D diffusion model, dreamscene2D, which is capable of generating high-fidelity and high-quality views of rooms following the rendered “bounding-box image” (containing 2D semantic maps and depth) from the bounding box representation. Dreamscene2D is fine-tuned from stable diffusion using LoRA adapters. This is then used for training a NeRF using Score Distillation Sampling. Additional measures, such as a depth constraint, floc removal and texture consolidation with backbone models helps generate the desired scene. In comparison with presented baselines, the results show the ability to model complex shapes in rooms, greater consistency across views, as well as the ability to generate objects of varying sizes. Ablation experiments are provided for the effect of the base prompt, the effect of the depth constraint and the effect of the texture consolidation using backbone models.

**Strengths:**

+ The paper is structured in a manner that is easy to understand
+ The method is reasonable, and the various components are well motivated in the text and through ablations.
+ In terms of the qualitative comparison with the shown baseline methods, clear differences can be seen with the proposed method leading to more detailed and accurate scenes
+ The authors promise to open-source the code post acceptance

**Weaknesses:**

- Lack of quantitative metrics: The paper claims superiority over baselines both qualitatively and quantitatively (L55). However, I am unable to find any quantitative metrics in the paper.
- Generation quality: While the proposed method seems to perform better than the chosen baselines, the quality of the generated scenes seems to be relatively lower overall. For example, in the last row of Figure 4, the scene lacks texture and shows a degeneracy in color generation.
- Lack of failure cases: In such works as scene generation, and especially since the scope of this work is limited to indoor scenes, I believe it is necessary to contextualize the kind of scenes where the proposed method is successful and where it fails.

**Questions:**

- Runtime: I believe the time taken for 3D asset generation through such text-guided approaches is an essential metric for comparison. However I am unable to find this quantification in this work.
- Quantitative metrics: I think it is important to quantify performance of the model (using maybe something like a CLIP score or FID), especially since this is mentioned in the introduction of the paper.
- It would be good to get an understanding of failure cases of the proposed method, or settings where the proposed method is unable to generate a favorable output.

For now, I would like to give a weak accept score and wait for the authors' responses to the mentioned questions and weaknesses.

**Limitations:**

The authors have addressed limitations and future directions in the appendix which is appreciated. I have some questions with regards to failure cases that I have addressed in previous sections.

---

> ### Author Rebuttal · Authors · 2024-08-07
>
> We appreciate your comments, and address your concerns as follows:
>
> ***
> 1. *Q: Evaluation: Authors should provide quantitative results.*
>
> We appreciate the reviewer's suggestion for quantitative results.
> * Our initial decision to prioritize qualitative evaluation was based on the challenges in defining fair, accurate, and comprehensive metrics for evaluating 3D scene generation methods across all aspects. The development of such metrics remains an open challenge in the generative AI community.
> * However, to address the reviewer's concern, we include quantitative comparison with baseline methods [11,24,55] on CLIP Score (CS), Inception Score (IS), and a user study on 3D scene consistency and overall visual quality. The experiment design follows previous work [48]. The detailed quantitative evaluation results are provided in **General Response**.
>
> ***
> 2. *Q: Overall quality: While the proposed method seems to perform better than the chosen baselines, the quality of the generated scenes seems to be relatively lower overall (especially Fig.4 second example).*
>
> We appreciate the reviewer's observation regarding the quality of generated scenes. We'd like to address this concern and highlight the strengths of our approach:
> * **Pioneering Complex Scene Generation**: Our primary contribution lies in achieving 3D-consistent complex scene generation with flexible layouts and free camera trajectories. We are the first to generate non-regular room shapes with complex layouts, as demonstrated by the L-shaped rooms in our supplementary Figure B and the new T-shaped 1-bedroom apartment scene provided in the rebuttal (PDF and video). This capability surpasses baseline models [16,48,55] that rely on panoramas as intermediate states and cannot achieve such non-regular room generation with precise layout control.
> * **Geometric Consistency**: While the second example in Figure 4 may have some limitations in color and texture, it maintains reasonable and consistent geometry that accurately follows the layout condition. This geometric fidelity is a crucial aspect where our method outperforms others (e.g., Text2Room, Set-the-Scene, MVDiffusion) that struggle to achieve comparable geometry quality or adhere to layout constraints.
> * **Improved results**: We have found that the quality of generated scenes can be significantly enhanced by using more stylish and thematic descriptions. In the rebuttal PDF, we've included two variants of the same layout with improved text prompts. Our experiments show that using stylish descriptions like "ocean-themed" or "Van Gogh style" yields much better results compared to using specific words like colors to define the scene.
>
> These improvements demonstrate the potential of our method to generate high-quality scenes when provided with appropriate prompts, while maintaining its unique capabilities in complex layout generation and geometric consistency.
>
> ***
> 3. *Q: Lack of failure cases: successful and failure cases should be provided and discussed.*
>
> We thank the reviewer for the suggestion. Here, we provide a summary over the common failure cases of our model. Analysis shows that two failure cases of our model are *Extremely Complicated Layout* and *Unmatched Prompts*. For more details and visualization please refer to the **General Response** and figures in our rebuttal PDF.
>
> ***
> 4. *Q: Training details: Lack of runtime data.*
>
> We thank the reviewer for the suggestion, and will clarify these points in our revised manuscript.
> * As described in suppl. Sec. A, we use a duo-GPU training scheduling.
>     * We use 2 A6000 GPU to conduct all our experiments and generally, each well-generated scene costs around 3-4 hours, the first GPU holds the process of diffusion model and the memory cost is ~6GB in FP16 with an image size of 512×768, another GPU holds the process of scene representation, and the memory cost is ~28GB (used Nerfacto from NerfStudio).
>     * For NeRF training, we use a constant learning rate of 1e-2 for proposal networks and 1e-3 for fields. We will add those to the main paper.
> * Note that some concurrent works cost more time, ShowRoom3D costs ~10h to produce a single scene (Tab.3 of their paper), UrbanArchitect costs ~12h and 32GB to produce a single scene (Sec.4.1 of their paper). Therefore, our DreamScene has higher efficiency than the baselines.

---

> > ### Comment · Area_Chair_bT2w · 2024-08-11
> >
> > Dear Reviewer,
> >
> > This is a gentle reminder to please review the rebuttal provided by the authors. Your feedback is crucial to the decision-making process. Please consider updating your score after reading the rebuttal.
> >
> > Thank you for your help with the NeurIPS!
> >
> > Best, Your AC

---

> > ### Comment · Reviewer_Nerf · 2024-08-13
> > **Post-Rebuttal Comment**
> >
> > I thank the reviewers for their careful response to the questions that were raised. I appreciate the presented quantitative results, the discussion on limitations and training details. Given the scope and visual quality of the results, and taking into account the author responses, I would like to retain my score of weak accept.

---

> > > ### Author Response · Authors · 2024-08-13
> > > **Thank reviewer Nerf for your positive feedback!**
> > >
> > > We appreciate the reviewer for the positive feedback. Your constructive comments and suggestions are indeed helpful for improving the paper.
> > >
> > > We will continue to improve our work and release the code. If the reviewer has any follow-up questions, we are happy to discuss!

---

### Official Review · Reviewer_DRMD · 2024-07-16

**Soundness:** 3
**Presentation:** 3
**Contribution:** 2
**Rating:** 6
**Confidence:** 4

**Summary:**

This paper addresses the task of 3D-consistent indoor scene generation using 2D diffusion models. As input the user provides a 3D bounding-box layout of the scene and a text prompt coarsely describing the scene. The proposed method contains a two-stage training phase. In the first stage, a 2D diffusion model is fine-tuned to synthesize a rendered view of a scene based on its rendered semantic layout and a depth map from the same view. In the second stage, the fine-tuned diffusion model is distilled into a NeRF representation with an SDS-based approach. The authors qualitative evaluate their method and compare with existing baselines, showcasing the effectiveness of their method.

**Strengths:**

- The paper is well-written and easy to follow.
- The proposed method is straightforward.
- The provided results show promising results compared to the baselines.
- The authors provide ablation study on their proposed components

**Weaknesses:**

- The proposed method might not be very novel. The contribution of the paper seems mainly to be fine-tuning the diffusion model for semantic-guided image synthesis. The distillation stage is mostly similar to the existing methods.

- The authors only evaluate their method qualitatively. Although the provided results seem very promising in terms of the 3D consistency of the scenes, a quantitate evaluation could provide more certainty about the advantage of the proposed method.

- The authors could have provided video visualizations in the supplementary, as assessing the 3D consistency of the method from multi-view images is challenging

**Questions:**

See the weaknesses

**Limitations:**

Adequately discussed.

---

> ### Author Rebuttal · Authors · 2024-08-07
>
> We appreciate your comments, and address your concerns as follows:
>
> ***
> 1. *Q: The proposed method might not be very novel. The contribution of the paper seems mainly to be fine-tuning the diffusion model for semantic-guided image synthesis. The distillation stage is mostly similar to the existing methods.*
>
> We would like to highlight the novelty and contributions of our work:
> * **Pioneering Free Multi-View Trajectories**: Our primary contribution lies in being the first to operate on free multi-view trajectories during scene generation. This approach contrasts sharply with baseline models [16,48,55], which rely on panoramas as intermediate states. As a result, we can now generate complex non-square rooms (e.g., L-shaped rooms, as demonstrated in supplementary Figure B and rebuttal PDF Figure.1), a capability beyond the reach of prior work.
> * Other than the novel non-panorama-based framework, our DreamScene has the following **technical contributions**, as also summarized in L58~L64:
>     * Bounding-Box Scene (BBS) Layout Condition. We introduce this novel layout condition, which offers greater versatility compared to layout conditions in previous work, inpainting-based approaches, and composition-based methods.
>     * We support text-conditioned generation to generate rooms with different styles, with our fine-tuned diffusion model which produces multiview semantic-rich guidance, which is not different from prior works using panorama guidance.
>     * We propose a novel NeRF distillation procedure that could remove the flocs via periodical migration and consolidate textures through unique loss design, as described in Sec. 3.3 Line 165-184. Additionally, we introduce multi-timestep sampling to enhance both semantic guidance and generation consistency.
> * Unlike existing layout-guided methods, we commit to releasing our code upon acceptance, fostering transparency and reproducibility in the research community.
>
> These contributions, particularly our non-panorama-based framework, represent significant advancements in the field of semantic-guided image synthesis and scene generation.
>
> ***
> 2. *Q: Evaluation: authors should provide quantitative results.*
>
> We appreciate the reviewer's suggestion for quantitative results.
> * Our initial decision to prioritize qualitative evaluation was based on the challenges in defining fair, accurate, and comprehensive metrics for evaluating 3D scene generation methods across all aspects. The development of such metrics remains an open challenge in the generation community.
> * However, to address the reviewer's concern, we include quantitative comparison with baseline methods [11,24,55] on CLIP Score (CS), Inception Score (IS), and a user study on 3D scene consistency and overall visual quality. The experiment design follows previous work [48]. The detailed quantitative evaluation results are provided in **General Response**.
>
> ***
> 3. *Q: Visualization: authors should provide videos to better show the 3d consistency in addition to multiview images.*
>
> Thank the reviewer for the suggestion. Please refer to **General Response** for details about our provided video visualization.

---

> > ### Comment · Area_Chair_bT2w · 2024-08-11
> >
> > Dear Reviewer,
> >
> > This is a gentle reminder to please review the rebuttal provided by the authors. Your feedback is crucial to the decision-making process. Please consider updating your score after reading the rebuttal.
> >
> > Thank you for your help with the NeurIPS!
> >
> > Best, Your AC

---

> > > ### Comment · Reviewer_DRMD · 2024-08-13
> > > **Post-Rebuttal Comment**
> > >
> > > I thank the authors for their answers, in particular for providing the quantitative evaluation and video visualisations. I raise my score to Weak Accept

---

> > > > ### Author Response · Authors · 2024-08-13
> > > > **Thank reviewer DRMD for your positive feedback!**
> > > >
> > > > We appreciate the reviewer for the positive feedback. Your constructive comments and suggestions are very helpful for improving the paper. Also, many thanks for raising the score.
> > > >
> > > > We will continue to improve our work and release the code. If the reviewer has any follow-up questions, we are happy to discuss!

---

### Author Rebuttal · Authors · 2024-08-07

# General Response

We greatly appreciate the thoughtful feedback and suggestions from all reviewers. We are pleased that the reviewers recognize the strengths of our approach, including more detailed and accurate scene generation (DRMD, Nerf), excellent controllability and user-friendly design (Gzhh, DHDm), and a well-structured, clearly written manuscript (DRMD, Nerf, DHDm).

**New Experiments and Clarifications**: We have addressed each reviewer's concerns individually. Below, we summarize the key experiments, visualizations, and clarifications added during the rebuttal phase.
* **Please refer to our attached PDF for all visualization figures.**
* **Video Visualization** (DRMD, DHDm): To demonstrate that our method generates scenes with complex shapes and layouts while maintaining strong 3D consistency, we provide videos of multiple generated scenes. We have emailed program chairs and got permission to provide links to anonymous videos in the rebuttal. **In compliance with the conference rules, we have sent the video to AC via a private link. Once AC has reviewed the video, it will be shared with all the reviewers.**
* **Additional Challenging Qualitative Results**(Gzhh): In addition to the video, we present more qualitative results of complex scenes in rebuttal Figure [1]. These results clearly illustrate our method's superior performance in terms of 3D consistency and complex scene generation compared to previous approaches.

* **Quantitative Results** (DRMD, Nerf, Gzhh, DHDm): In our original submission, we prioritized qualitative evaluation for various types and styles of generation, as we found it challenging to define fair, accurate, and comprehensive metrics for quantitative evaluation of different 3D scene generation methods.
    * However, to address the reviewers' concerns, we now include quantitative comparisons with baseline methods [11,24,55] using CLIP Score (CS) and Inception Score (IS). We also conducted a user study with 37 participants, who scored the 3D consistency and overall visual quality of rooms generated by different methods on a scale of 1 to 5. Our experimental design follows previous work [48].
    * The quantitative results are presented in the table below. Our method consistently outperforms prior approaches in terms of CLIP Score, 3D Consistency, and Visual Quality. Regarding the Inception Score, our diffusion model's finetuning with fixed categories somewhat limits generation diversity. However, this is not a major concern for our task, as previous works struggle to achieve both high consistency and visual quality while being controlled by layout prompts.

| **Method**        | CS ↑      | IS↑      | Consistency ↑ | Visual Quality ↑ |
|:-----------------:|:---------:|:--------:|:-------------:|:----------------:|
| Text2Room         | 22.98     | 4.20     | 3.12          | 3.08             |
| MVDiffusion       | 23.85     | **4.36** | 3.20          | 3.47             |
| Set-the-scene     | 21.32     | 2.98     | 3.58          | 2.45             |
| **DreamScene** (Ours) | **24.34** | 3.54     | **3.91**      | **3.68**         |


* **Discussion on Failure Cases** (Nerf, DHDm): We discuss and clarify the failure cases and limitations of our method.
    * **Extremely Complicated Scenes**: Our method may struggle to reason the layout and generate rooms when the layout is excessively complex, containing many closely-placed objects or highly overlapped bounding boxes (see Figure [2] in our rebuttal PDF).
    * **Mismatched Prompts**: When the prompt does not align with the actual room layout (e.g., a bedroom layout with a "kitchen" prompt), our method may fail to generate appropriate room contents or achieve convergence (see Figure [3] in our rebuttal PDF).
* **Improved Visual Quality** (Nerf, Gzhh): We demonstrate that generation quality is closely tied to prompt control. Our experiments reveal that using stylistic descriptions (e.g., "ocean-themed", "Van Gogh style") leads to greater quality improvements compared to specific descriptors like colors. We can further optimize appearance quality by refining textual prompts, as illustrated in rebuttal Figure [4].
* **Training Details** (Nerf, Gzhh, DHDm): We provide comprehensive training details for our model, including runtime, memory usage, and other relevant parameters used in our experiments. Our code will be made publicly available upon acceptance of the paper.
    * As described in suppl. Sec. A, we use a duo-GPU training scheduling.
        * We use 2 A6000 GPU to conduct all our experiments and generally, each well-generated scene costs around 3-4 hours, the first GPU holds the process of diffusion model and the memory cost is ~6GB in FP16 with an image size of 512×768, another GPU holds the process of scene representation, and the memory cost is ~28GB (used Nerfacto from NerfStudio).
        * For NeRF training, we use a constant learning rate of 1e-2 for proposal networks and 1e-3 for fields. We will add those to the main paper.
    * Note that some concurrent works cost more time, ShowRoom3D costs ~10h to produce a single scene (Tab.3 of their paper), UrbanArchitect costs ~12h and 32GB to produce a single scene (Sec.4.1 of their paper). Therefore, our DreamScene has higher efficiency than the baselines.

---

> ### Author Response · Authors · 2024-08-09
> **Video Link**
>
> Dear Reviewers, here is the **link to the anonymous video with our generated scenes**. https://github.com/DreamScene3D/DreamScene/raw/main/rebuttal_video.mp4

---

> > ### Comment · Reviewer_Gzhh · 2024-08-12
> > **Clarification required for the quantiative results**
> >
> > I appreciate the authors for the dedicated response. I have some questions regarding the provided quantiative metrics.
> > Is the quantiative metrics presented here during the rebuttal with same datasets? In the rebuttal to the comments that I have wrote, the authors mentioned that
> >
> > > Our diffusion model is primarily fine-tuned on the synthetic Hypersim [45] dataset, chosen for its high-quality layout data. This choice aligns with other recent works like ControlRoom3D [48].
> >     In contrast, MVDiffusion uses a version of Stable Diffusion fine-tuned on the real-world ScanNet dataset, while Text2Room uses the original Stable Diffusion model.
> >
> > which means that models to obtain the quantiative metrics might have been trained with different datasets, as far as I know.
> > Could the authors clarify this? Because, the dataset matters and might be critical to the obtained quantitative results. Also, what were the choices for not using FID as an evaluation metric?

---

> > > ### Author Response · Authors · 2024-08-13
> > > **Reply to Reviewer Gzhh's feedback at General Response**
> > >
> > > We sincerely appreciate reviewer *Gzhh* for checking our rebuttal. We address this remaining concern as below.
> > >
> > > ***
> > > 1. **Different training datasets used for quantitative evaluation.**
> > >     - Our DreamScene2D is designed to generate high-quality 3D scenes that match the input layout. To achieve this, we require a dataset with **high-quality 3D bounding box annotations** with accurate category labels for training.
> > >         - We therefore chose to train our DreamScene2D using the Hypersim [45] dataset. This choice of the training dataset is also *consistent* with ControlRoom3D [48].
> > >         - Other baseline methods use ScanNet as their training dataset. However, the bounding box annotations in ScanNet are of low quality (noisy and inaccurate), and cannot satisfy our requirements.
> > >         - While we understand the reviewer’s concern that using different training datasets might influence specific quantitative metrics, re-training the baseline models using the Hypersim dataset we employ is challenging and time-consuming. Additionally, we think that dataset selection is a critical design choice in developing these methods and should be considered during evaluation. For example, as mentioned above, we chose Hypersim for DreamScene2D specifically because of its high-quality 3D bounding box annotations. Therefore, using the exact same training data for comparison in our context may not be practical or necessary.
> > >         - In fact, comparing methods trained on different datasets is a common practice in generative model research. For example, ControlRoom3D [48] also compares with methods (MVDiffusion and Text2Room) trained on different datasets. Similarly, the well-known 2D diffusion model Instruct-Pix2Pix is compared with T2L and SDEdit, which are trained on other datasets.
> > >     - Importantly, our quantitative experiments focus on the CS, IS, and user study metrics, which evaluate the final generation results using either pre-trained models (CLIP and Inception) or participants’ judgments. These metrics do *not* require ground-truth scenes from the dataset and, therefore, are *agnostic* to the dataset.
> > >     - Therefore, even when using a different dataset for training, our quantitative comparison remains fair and reasonable with the current metrics and adheres to established practices.
> > >
> > > 2. **Not using FID metrics.**
> > >     - First, both our method and [48] focus on the overall quality of generated scenes and layout-following capability, unlike MVDiffusion which focuses on the quality of the diffusion image generation. Therefore, our method and [48] choose to report current metrics rather than FID.
> > >     - On the other hand, the value of the FID metric is dependent on the dataset. Since we use different datasets for diffusion finetuning, applying FID would result in unfair and inaccurate comparison.
> > >
> > > ***
> > > If the reviewer has any follow-up questions, we are happy to discuss them.

---

### Decision · Program_Chairs · 2024-09-25

**Decision:**

Accept (poster)

**Comment:**

This paper introduces an interesting approach to generating scenes given the 3D layout. The reviewers questioned technical novelty, evaluation of the 3D consistency, more visualization, training details, and limitation analysis during the review phase. The authors provided solid feedback to relieve the concerns, and most of the reviewers mentioned that the rebuttal helps to understand the methods better, and the provided additional results are acceptable. After the rebuttal phase, reviewer DRMD raised the score to 'weak accept', reviewer Nerf keep the 'weak accept' score, reviewer Gzhh increased the score to weak accept, and reviewer DHDM keep the score 'borderline accept'. As a result, no reviewers opposed the paper's rejection. Although there is a concern about the similarity of the concurrent work published in ArXiv, AC confirmed that that issue could not affect paper judgment, given the NeurIPS paper submission policy. AC also confirms that the paper has several merits worth presenting in the field.